# Modeling the Effects of Local Atmospheric Conditions on the Thermodynamics of Sobradinho Lake, Northeast Brazil

Eliseu Oliveira Afonso *,† and Sin Chan Chou †

National Institute for Space Research-INPE, Rodovia Presidente Dutra, km 40 SP/RJ, Cachoeira Paulista, São Paulo CEP 12630-970, Brazil; chou.chan@inpe.br

* Correspondence: eliseuoafonso@gmail.com
† These authors contributed equally to this work.

**Abstract:** The objective of this work was to study climate variability and its impacts on the temperature of Sobradinho Lake in Northeast Brazil. Surface weather station data and lake measurements were used in this study. The model applied in this work is FLake, which is a one-dimensional model used to simulate the vertical temperature profile of freshwater lakes. First, the climate variability around Sobradinho Lake was analyzed. Observations showed a reduction in precipitation during 1991–2020 compared to 1981–2010. To study climate variability impacts on Sobradinho Lake, the years 2013, 2015, and 2020 were selected to characterize normal, dry, and rainy years, respectively. In addition, the months of January, April, July, and October were analyzed for rainy months, rainy–dry transitions, dry months, and dry–rainy transitions. Dry years showed higher incoming solar radiation at the surface and, consequently, higher 2 m air temperatures. A characteristic of the normal years was more intense surface winds. October presented the highest incoming solar radiation, the highest air temperature, and the most intense winds at the surface. The lowest incoming solar radiation at the surface was observed in January, and the lightest wind was observed in April. To assess the effects of these atmospheric conditions on the thermodynamics of Sobradinho Lake, the FLake model was forced using station observation data. The thermal amplitude of the lake surface temperature (LST) varied by less than 1 °C during the four months. This result was validated against surface lake observations. FLake was able to accurately reproduce the diurnal cycle variation in sensible heat fluxes (H), latent heat fluxes, and momentum fluxes. The sensible heat flux depends directly on the difference between the LST and the air temperature. During daytime, however, Flake simulated negative values of H, and during nighttime, positive values. The highest values of latent heat flux were simulated during the day, with the maximum value was simulated at 12:00 noon. The momentum flux simulated a similar pattern, with the maximum values simulated during the day and the minimum values during the night. The FLake model also simulated the deepest mixing layer in the months of July and October. However, our results have limitations due to the lack of observed data to validate the simulations.

**Keywords:** lake surface temperature; FLake model; Sobradinho Lake





## 1. Introduction

Sobradinho Lake is located along the São Francisco River, Brazil, centered at approximately latitude −9.67° and longitude −41.50°. It is considered one of the largest artificial lakes in the world. The lake was built mainly to regulate water flow from the São Francisco River [1]. The major activities related to the lake are energy production, crop irrigation, and fishing.

Sediment retention behind power generation dams, however, significantly impacts water turbidity, disrupting fish physiology. Decreases in water temperature, especially in the deeper parts of the reservoir, may cause disturbances in the reproduction and development of some fish species [2].

Sobradinho Lake has also affected the region's atmospheric weather and climate conditions (Correia et al. [3]). Campos [4], using a space–time analysis of climate variability, evaluated the influence of Lake Sobradinho on the surrounding climate. The author found a 13% increase in precipitation in the cities around the lake and a 16% increase in precipitation in the wettest quarter.

Using a Regional Atmospheric Modeling System model to simulate the climate around the Sobradinho Lake area, Sobral et al. [5] verified a direct link between variations in reservoir levels and significant changes in climate variables. The authors also showed that the topographic features of the lake area contributed to the development of a complex circulation system, including lake breeze, land breeze, anabatic wind, and katabatic wind. However, changes in lake size and geometry alter the spatial distribution of lake-breeze-induced convergence zones, producing diurnal variations in climatic elements.

Ekhtiari et al. [6] analyzed the effects of Sobradinho Lake on the regional climate by coupling the one-dimensional FLake (freshwater lake) model with the COSMO-CLM regional model [7] and comparing a simulation with and without the presence of the lake. However, the lake was replaced by the surrounding vegetation. The authors identified cooling in the morning and heating at night of up to 3 °C in the air around the lake, together with a significant increase in the relative humidity.

Ultimately, numerical simulations using lake models can help capture the lake's physical processes and their influences on the local and regional climate. The interactions between the atmosphere and the physics and dynamics of lakes deserve further investigation, including validation of the lake model simulation [8]. By using a lake model, this work aimed to study climate variability around Sobradinho Lake and its influence on lake temperature variability.

This article is organized as follows. The methodology is described in Section 2. Section 3 contains a description of the climate mean and its variability around Sobradinho, an analysis of the variability in surface atmospheric variables during the last 30 years, and a discussion of effects on lake temperatures and energy fluxes, simulated by the FLake model. Finally, conclusions are provided in Section 4.

## 2. Materials and Methods

### 2.1. Study Area

The study area is the Sobradinho Lake, located in Northeast Brazil (NEB) (Figure 1). This lake extends for about 320 km and features a water surface of 4214 km$^2$, a capacity of 34.1 billion cubic meters, and a regulated flow of 2060 m$^3$/s during dry periods.

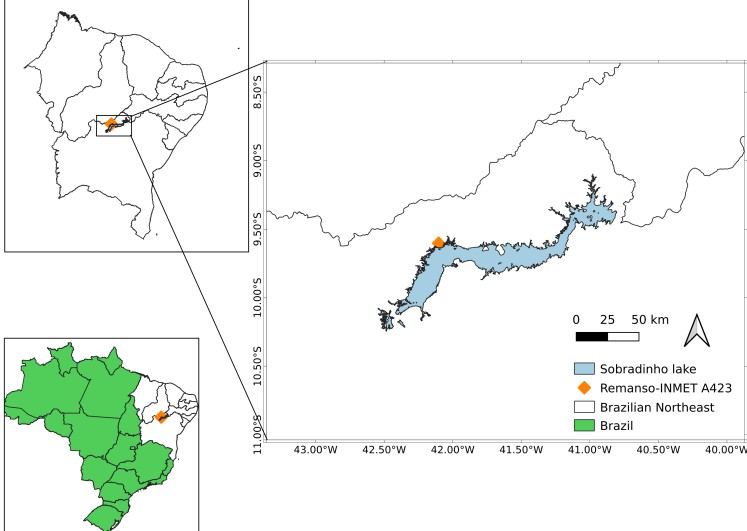

**Figure 1.** Study area: Sobradinho Lake is marked in blue, and the orange dot is the meteorological station Remanso.

### 2.2. Observed Data

The observed data were obtained from the Brazilian National Institute of Meteorology (INMET) weather station at latitude $-9.62°$ and longitude $-42.07°$. The conventional station contains data from 1 January 1961, with a daily frequency, while automatic station data are available starting from 25 March 2008.

The conventional weather station reports with a daily frequency and its long-time series were used to describe the surrounding climate. The automatic weather station reports hourly, and its data were used to describe the diurnal cycle of meteorological variables and as an initial condition for the FLake model. The stations report precipitation, 2 m air temperature, 10 m surface wind, and surface incident solar radiation.

Observations from January (rainy month), April (rainy–dry transition month), July (dry month), and October (dry–rainy transition month) during the years 2013, 2015, and 2020 were used in this study. These years were selected to show interannual variability and represent a climatologically normal year, a dry year, and a rainy year, respectively. The four months capture some intraseasonal variability in the observational data.

### 2.3. FLake Model

The FLake model [9] was used to simulate the lake conditions of Sobradinho Lake. FLake is a two-layer model that simulates the thermodynamic characteristics of small and large lakes and is capable of predicting the vertical temperature structure, mixing conditions, and surface turbulent fluxes in lakes of various depths in time scales from a few hours to many years ([10,11]). The FLake model can be used alone or coupled with weather and climate forecast models.

Twelve model runs were used to simulate January, April, July, and October for 2013, 2015, and 2020. The numerical simulations were calibrated with the atmospheric characteristics of a tropical region with no ice formation or snow accumulation.

Atmospheric forcing variables were entered at an hourly frequency, and the lake depth was set to 20 m, as this value is regularly used in measurements by the Department of Fisheries of the Federal Rural University of Pernambuco (DEPESCA/UFRPE). However, the lake depth can vary up to a maximum of 30 m. The extinction coefficient, water albedo, and fetch length were set using values defined by Mironov [12] (Table 1).

**Table 1.** FLake model parameters used in this study.

| Parameters | Value |
| --- | --- |
| Depth | 20 m |
| Fetch | 12,500 m |
| Extinction coefficient | $3\ \text{m}^{-1}$ |
| Water albedo | 0.07 |
| Timestep | 3600 s |

## 3. Results

This section analyzes the precipitation regime, variability of incident solar radiation, air temperature at 2 m, and wind magnitude at 10 m. Precipitation data characterized the dry, normal, and rainy years. Solar radiation, air temperature, and wind magnitude were analyzed as input variables for the FLake model. Subsequently, the impacts of atmospheric variables on the lake surface temperature, lake stratification, and energy fluxes were evaluated.

### 3.1. Interannual Variability

The interannual variability in precipitation anomalies between 1991 and 2020 (Figure 2) shows that one standard deviation varies between 200 and −200 mm. The average of the annual accumulated precipitation was 607.90 mm. Here, dry and rainy years are defined as years of precipitation above or below one standard deviation, respectively, relative to years with normal rainfall.

The observations indicated 20 normal years, 4 rainy years, 6 dry years, and 1 (1993) extremely dry year. Additionally, 1993 and 2015 were characterized as drought years, and both years were related to episodes of weak and strong El Niño, respectively. In contrast, 2000 was considered an extremely rainy year and related to a La Niña episode [13].

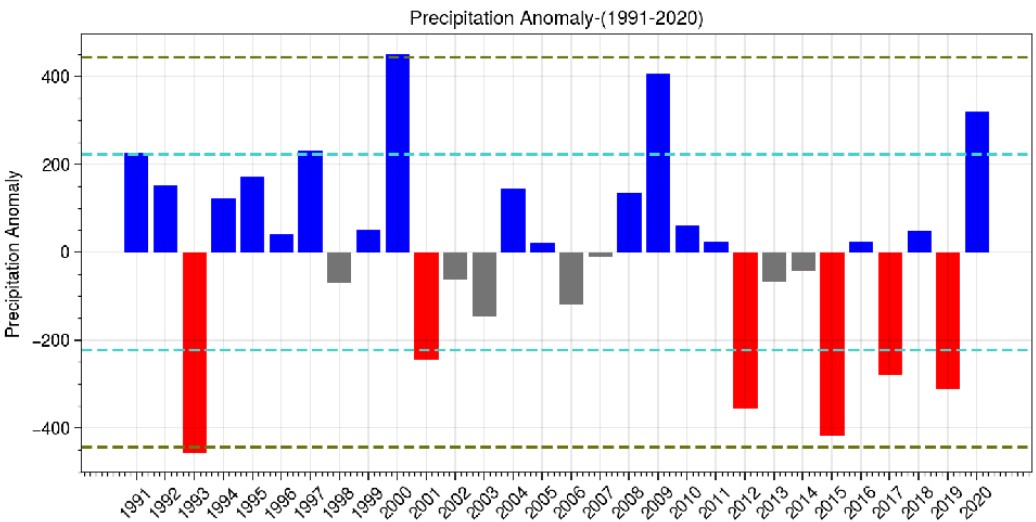

**Figure 2.** Annual precipitation anomaly. Wet years are shaded in blue, normal years in gray, and dry years in red. A year is defined as normal when the precipitation anomaly is within one standard deviation (light blue dashed line). Extreme years are identified when the anomaly exceeds two standard deviations (gray dashed lines).

Figure 3 shows the interannual variability of incoming solar radiation, air temperature, and wind near the surface from 2009 to 2020. The year 2016 presented the highest incoming solar radiation (1020 W/m$^2$), followed by 2011 and 2014 (985 and 980 W/m$^2$, respectively). The first quarter of 2016 experienced strong El Niño conditions and accumulated four previous consecutive years of annual precipitation below average. This lack of precipitation caused severe drought in Northeastern Brazil [14]. The warmest years were 2015 and 2019, with an average air temperature of around 28 °C. The period 2013–2015 presented the strongest winds and were dry years, whereas 2009 and 2020 experienced weak winds and were also the wettest.

The precipitation regime in Northeast Brazil depends directly on the intensity of the southeasterly trade winds. In dry years, such as 2015, the trade winds are more intense and cause the northward displacement of the intertropical convergence zone (ITCZ). During dry years, the above-normal pressures of the South Atlantic subtropical high (SASH) intensify the southeasterly trade winds. On the other hand, in rainy years, the ITCZ is positioned more southward due to weakening of the SASH, which consequently weakens the southeasterly trade winds [15].

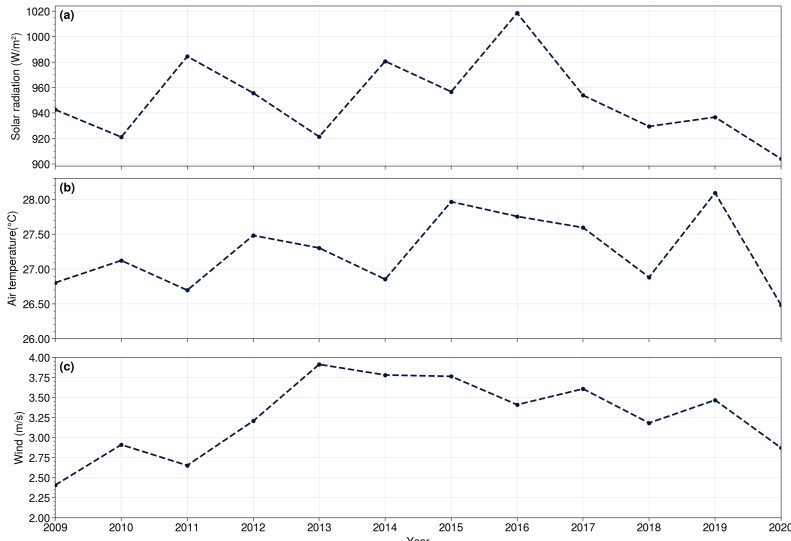

**Figure 3.** Interannual variability of (**a**) incoming solar radiation at the surface (W/m²), (**b**) 2 m air temperature (°C), and (**c**) 10 m wind speed (m/s).

### 3.2. Annual Cycle

The monthly precipitation in Sobradinho Lake shows well-defined dry and rainy seasons. The rainy season starts in November and ends in April, so the dry season starts in May and ends in October (Figure 4). March is the wettest month, with monthly precipitation of about 364.9 mm, and August is the driest, with monthly precipitation of about 1 mm. Figure 4 shows the average taken from two periods, one from 1981 to 2010 and the other from 1991 to 2020. In the climatology of 1981–2010, during the rainiest month, accumulated precipitation totaled about 414.5 mm, whereas, in the updated climatology of 1991–2020, the total precipitation during the rainiest month was reduced by about 49.6 mm ([16,17]). The transitional months of May and October also showed reductions in the monthly precipitation from the 1981–2010 climatological period to the 1991–2020 period. In May, the reduction totaled 23.80 mm to 18.35 mm, and in October, the reduction was 60.34 mm to 55.05 mm, thus decreasing annual total precipitation. The standard deviation decreased from the first to the second period. This decrease shows some changes in the climatological precipitation distribution over the years. Therefore, the lake's hydrological year is considered to span from November 1 to October 31. In addition, despite the two climatological periods overlapping by some years, there are changes in the statistics indicating local climate change.

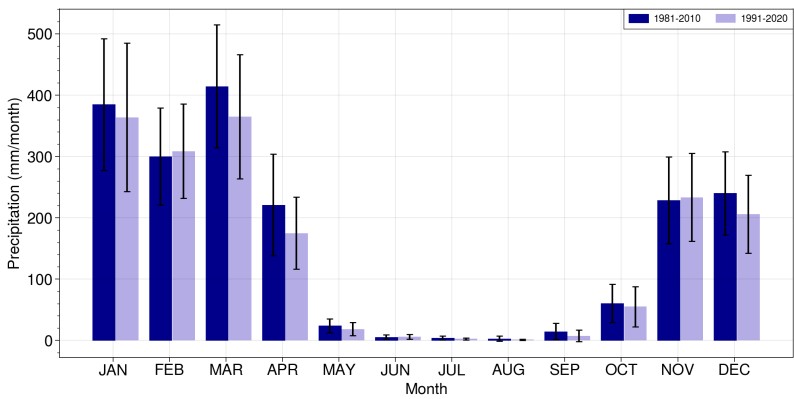

**Figure 4.** Monthly precipitation (mm/month) averaged for the periods 1981–2010 (dark navy) and 1991–2020 (light navy). Vertical bars indicate to the standard deviation.

Regarding the annual cycle (Figure 5), the maximum incoming solar radiation was observed in September and reached about 1100 W/m$^2$. In contrast, the minimum incoming solar radiation of about 820 W/m$^2$ was observed in June (austral winter). The temperature peaked in October and was lowest in July. Therefore, the temperature cycle lagged behind the incoming solar radiation cycle by about one month. The annual cycle of wind speed experienced monthly variation following weakening and strengthening of the southeasterly trade winds. The winds were weakest in the rainiest month and strongest toward the end of the dry season.

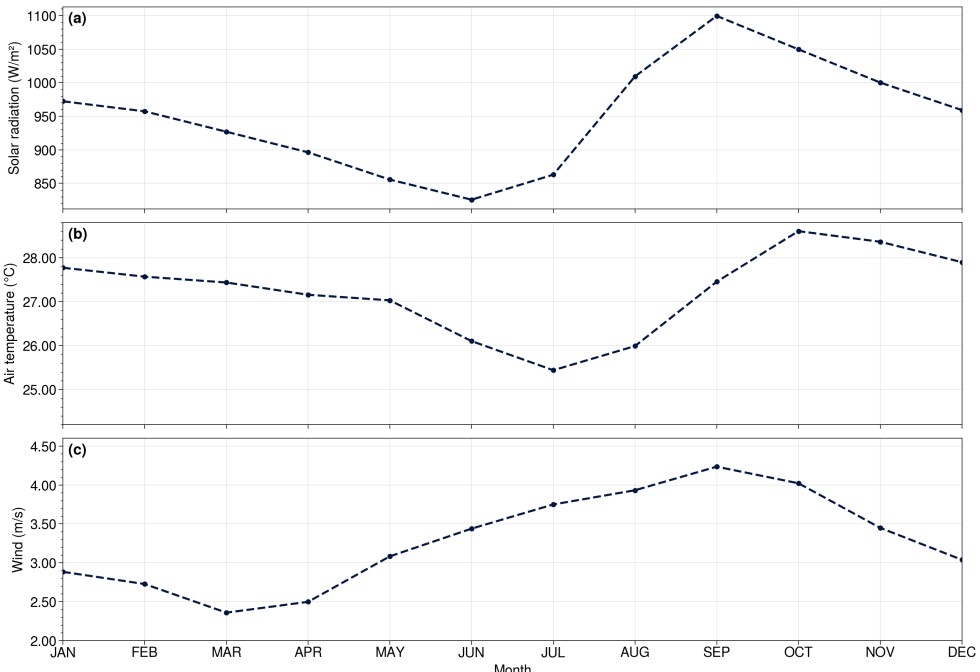

**Figure 5.** Mean annual cycle of (**a**) incoming solar radiation (W/m$^2$), (**b**) 2 m air temperature (°C), and (**c**) 10 m wind speed (m/s). The average is calculated for 2009–2020.

### 3.3. Atmospheric Forcings

The atmospheric forcings considered here are atmospheric variables used as input to drive the lake model, including incoming solar radiation at the surface, air temperature, and wind speed near the surface. These variables were analyzed hourly to investigate the diurnal cycle and interannual variability.

Figure 6 shows the average for the solar radiation diurnal cycle during 2009–2020. The diurnal cycle of the incoming solar radiation at the surface during dry, normal, and rainy years shows that the most significant amplitude occurred in October, reaching 400 W/m$^2$ (Figure 6d). January had the second-highest values (Figure 6a).

In April and July, the incoming solar radiation at the surface decreased to about 350 W/m$^2$. When the maximum radiative fluxes occurred in the northern hemisphere, the results for July were slightly lower than those for April (Figure 6b,c). The incoming solar radiation at the surface in January indicates sensitivity to interannual variability. The fluxes of the peaks varied from 390 to 310 W/m$^2$ during dry and rainy years, respectively. In January, more clouds occurred under normal and rainy years, and the incoming solar radiation was much lower than that during dry years. The peak of incoming solar radiation during dry January months may approach the radiation peak in October.

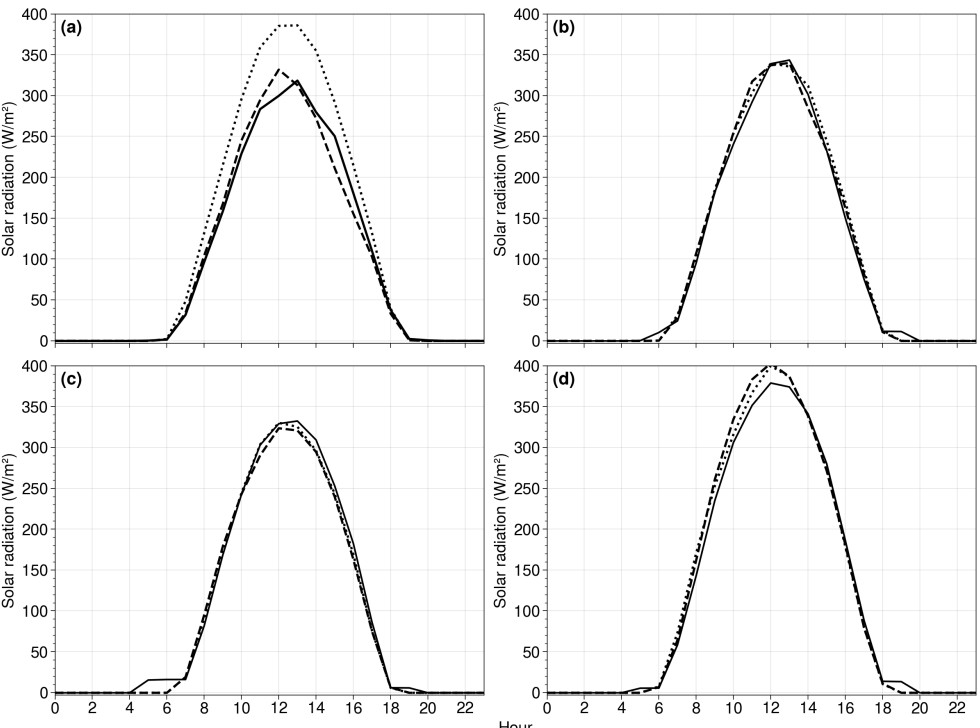

**Figure 6.** Diurnal cycle of incoming solar radiation at the surface (W/m²) for (**a**) January, (**b**) April, (**c**) July, and (**d**) October. The normal year (dashed lines) is 2013, the dry year (dotted lines) is 2015, and the rainy year (solid lines) is 2020.

The 2 m air temperature shows a diurnal amplitude varying from 6.00 to 10.00 °C (Figure 7). The rainy year shows the smallest amplitude at about 6.00 °C. In contrast, during the dry year, the largest amplitude reached about 9.00 °C. This difference in the rainy year is due to the greater presence of clouds, which prevented much of the direct solar radiation from reaching the Earth's surface and absorbing the long-wave radiation emitted by the Earth's surface, making the daytime temperature cooler. The nighttime was, consequently, warmer, which yields a smaller thermal amplitude. On the other hand, there is little cloudiness during dry years, making the daytime warmer and the nighttime cooler. In July, the minimum temperature reached 21.00 °C. During October for the three years under study, the maximum temperature reached 34.00 °C in the dry year and 33.00 °C in the normal and rainy years. Therefore, October is the warmest month during the period under study (Figure 7d).

The wind was weaker during the rainy months due to weakening of the southeasterly trade winds, and the wind was stronger in the dry period due to strengthening of the trade winds (Figure 8). During the rainy year, the wind magnitude at 10 m was observed to reach a maximum of 5.00 m/s and a minimum of 1.40 m/s. In the dry year, the maximum reached 6.60 m/s. Generally, October presented the strongest winds. We also observed that the dry year had a weaker wind magnitude at 10 m than during the normal year, especially in October (Figure 8d). We expected the winds to be stronger during the dry year. However, this dry year was also an El Niño year. As part of this phenomenon, the Walker circulation cell is split into two branches, with one of the descending branches causing subsidence in the northern region of South America and, consequently, weakening the trade winds [18].

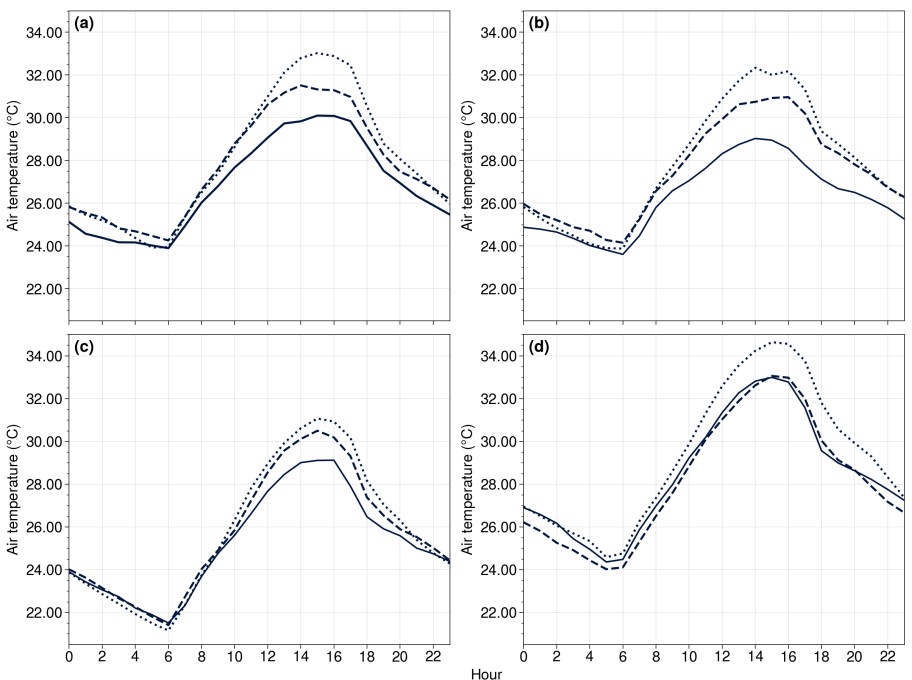

**Figure 7.** Diurnal cycle of 2 m air temperature (°C) for (**a**) January, (**b**) April, (**c**) July, and (**d**) October. The normal year (dashed lines) is 2013, the dry year (dotted lines) is 2015, and the rainy year (solid lines) is 2020.

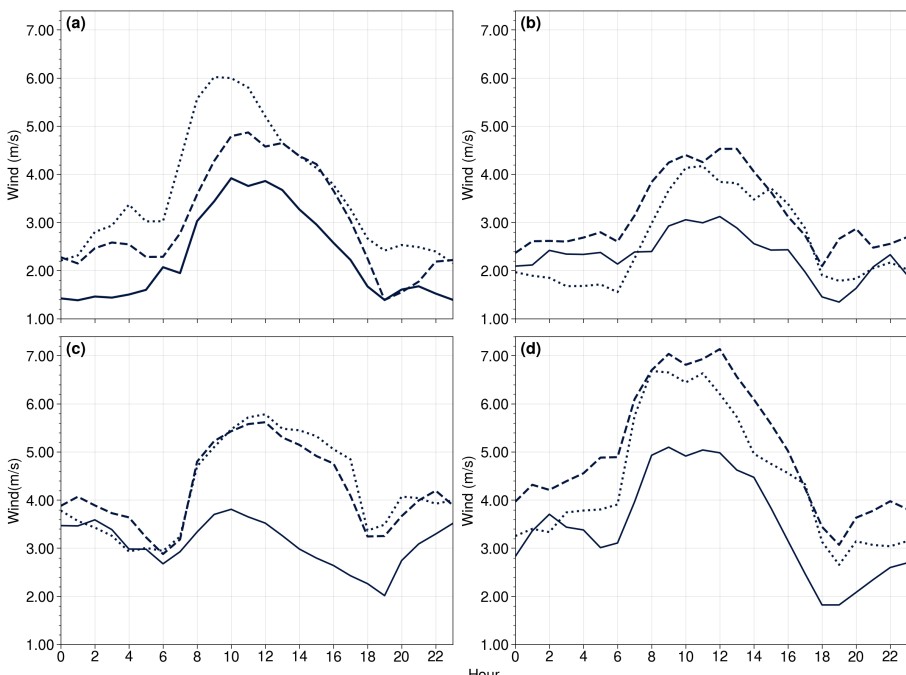

**Figure 8.** Diurnal cycle of 10 m wind speed (m/s) for (**a**) January, (**b**) April, (**c**) July, and (**d**) October. The normal year (dashed lines) is 2013, the dry year (dotted lines) is 2015, and the rainy year (solid lines) is 2020.

*3.4. Lake Conditions*

The FLake Model was applied to simulate Sobradinho Lake under different atmospheric forcing conditions. The diurnal cycle of the lake surface temperature (LST), mixing layer depth, and energy fluxes were investigated, which were the output variables used in the FLake model.

### 3.4.1. Lake Surface Temperature

The simulations showed a thermal amplitude of up to 1 °C in the four months. In the diurnal cycle, the LST increased starting from 6:00 a.m. local time, with a peak at 1:00 p.m. This behavior occurred due to diurnal variation in the incident solar radiation heating the lake's surface.

In January, the model showed an LST variation of 27.20–27.77 °C in the dry year, 27.22–27.71 °C in the normal year, and 27.28–27.72 °C in the rainy year. The dry season experienced a higher incidence of solar radiation (Figure 9) and, consequently, a higher LST thermal amplitude.

In April, the model showed the highest LST, whereas in the dry year, the simulations presented variations of 28.58 °C for the minimum and 29.09 °C for the maximum. In the normal and rainy years, the model presented the same minimum LST (28.56 °C), and the maxima were 29.04 °C (normal year) and 29.12 °C (rainy year). In this case, during the rainy year, LST had a higher maximum because, in addition to incident solar radiation, wind also plays an important role in LST variation. Rain helps to redistribute heat and vary the LST since it promotes turbulence in the surface layer, in addition to the presence or absence of layers of water with different densities ([19,20]). Consequently, the maximum LST was higher in the rainy year with weaker wind than in the dry and normal years.

In July, when the maximum did not exceed 25.50 °C, the model presented the lowest LST. In the dry and normal years, the model showed similar behavior for LST variation, with a minimum of 24.90 °C and a maximum of 25.38 °C. In the rainy year, the model instead indicated 25.00 °C as the minimum and 25.48 °C as the maximum.

In October, the model presented similarities between the minimum and maximum LSTs in the dry and rainy years, with the minimum reaching 26.20 °C and a maximum of 26.45 °C. In the normal year, the model indicated a minimum of 26.10 °C and a maximum of 26.42 °C.

### 3.4.2. Turbulent Energy Fluxes at the Surface

Energy fluxes are fundamental to the energy balance of the lacustrine system. Sensible heat flux (H) is related to adding or removing energy from fluid molecules during the heating or cooling process without any state change. A negative H value indicates the addition of heat to the lake surface, and a positive H value indicates the removal of heat from the lake surface to the surrounding atmosphere.

Latent heat flux (LE) is related to evaporation or condensation caused by turbulent heat transfer. Therefore, a high LE indicates heat transferred by the lake for the evaporation process, while a low LE indicates heat gained by the lake due to the transport of condensed water mass. For the momentum flux, the higher the wind speed, the greater the mechanically generated turbulent mixing in the lake ([21,22]).

In the simulation of H, the model showed the lowest H values (negative) on the lake's surface during January and April, reaching $-32.00$ W/m$^2$ in January and $-20.00$ W/m$^2$ in April, at least during the dry year. In the normal year, the values were $-30.00$ and $-16.00$ W/m$^2$, and in the rainy year, the values were $-14.00$ and $-2.00$ W/m$^2$ in January and April, respectively (Figure 10a,b).

In the rainy year, due to the low incidence of solar radiation and, consequently, low thermal amplitude, the H values were lower than those during the dry and normal years. In July, the negative H values on the lake's surface were as high as $-38.00$ W/m$^2$ in the dry year, $-34.00$ W/m$^2$ in the normal year, and $-14.00$ W/m$^2$ in the rainy year.

The highest H occurred in October, whereas, during the dry year, the negative H values were $-48$ and $-47.00$ W/m$^2$ in the normal year and $-36.00$ W/m$^2$ during the rainy year. These variations in H by the lake's surface, both negative and positive, show that when negative H values are more significant than those that are positive, the lake is gaining more energy, i.e., the lake is in a heating phase. When the opposite occurs, as in April of the rainy year (Figure 10b), the lake absorbs less and emits more thermal energy.

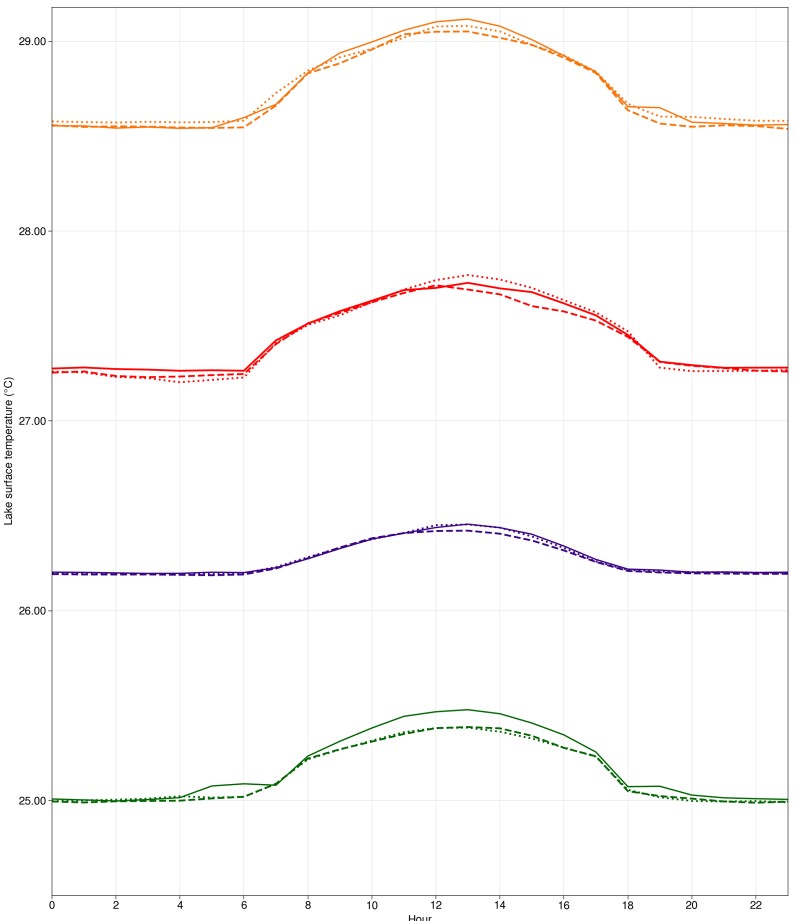

**Figure 9.** Diurnal cycle of lake surface temperature (°C). Dry years are dotted lines, normal years are dashed lines, and rainy years are solid lines. Red lines refer to January, orange lines to April, green lines to July, and purple lines to October.

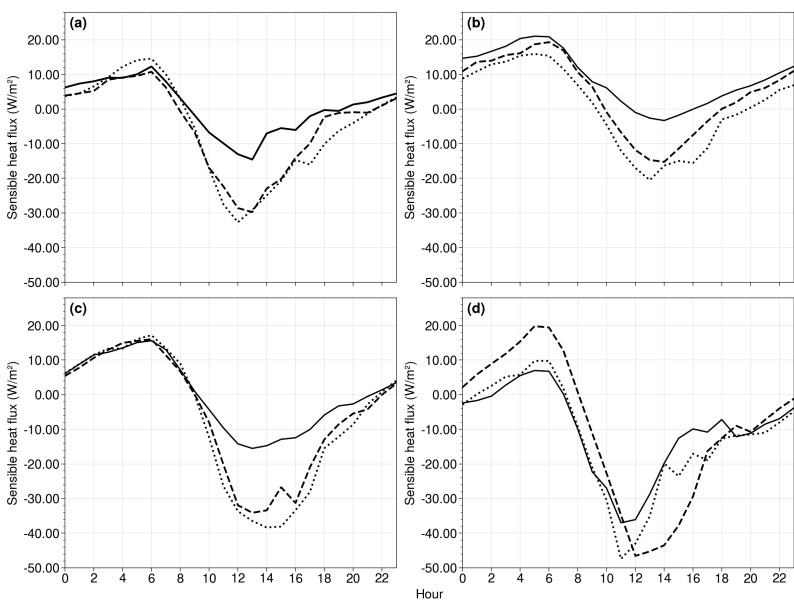

**Figure 10.** Diurnal cycle of sensible heat flux (W/m$^2$) for (**a**) January, (**b**) April, (**c**) July, and (**d**) October. The normal year (dashed lines) is 2013, the dry year (dotted lines) is 2015, and the rainy year (solid lines) is 2020. Positive values refer to upward fluxes out of the lake, and negative values refer to downward fluxes into the lake.

Figure 11 shows the LE diurnal cycle simulated by FLake. This model reproduces the LE daily cycle, in which the minimum values were obtained for the night period, and the maximum values were obtained for the day. In January, the dry year had higher LE values than those during the normal and rainy years. This difference occurs because during the dry year, the magnitude of the wind was more significant than during the normal and rainy years, as wind is a fundamental variable in the evaporation process. During this process, water molecules are accumulated at the lake–atmosphere interface, creating a layer of saturated air. Through the action of the wind, this layer is moved from the liquid surface and mixes with atmosphere. Therefore, the more intense the wind becomes, the more water vapor is drawn into the atmosphere ([23,24]). In the rainy year, due to the low incidence of solar radiation at the surface caused by cloudiness and weaker winds, especially in April (Figure 11b), the LE reduced to a maximum of 299.00 W/m$^2$.

In July, the dry and normal years showed similar patterns in the LE, reaching a maximum of 440.00 W/m$^2$. In October, the model yielded a higher LE during the three years under analysis, mainly because October is the month with the highest wind intensity. In addition, in the normal year, there was a slight increase in the LE compared to the dry year. As shown in Figure 11d, the magnitude of the wind during October in the normal year was more significant than in the dry and rainy years.

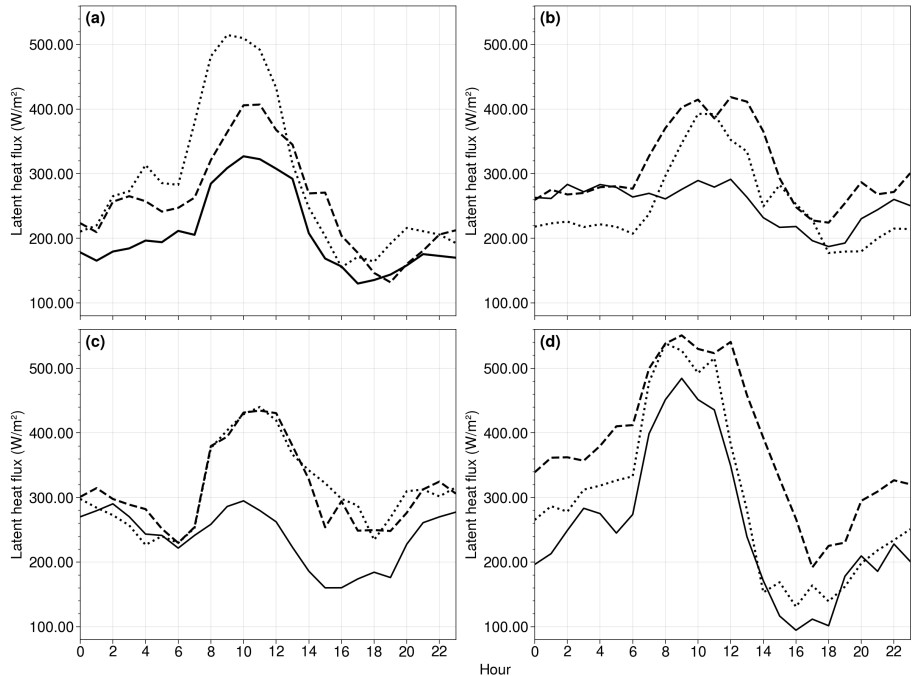

**Figure 11.** Diurnal cycle of latent heat flux (W/m$^2$) for (**a**) January, (**b**) April, (**c**) July, and (**d**) October. The normal year (dashed lines) is 2013, the dry year (dotted lines) is 2015, and the rainy year (solid lines) is 2020. Positive values refer to upward fluxes out of the lake, and negative values refer to downward fluxes into the lake.

The momentum flux is directly linked to wind intensity, thereby causing turbulent and mixing processes at the lake–atmosphere interface. Therefore, in periods of the day with more intense wind, higher momentum flux can be generated. The opposite occurs for weaker winds. In January, during the dry year, higher momentum flux was simulated in the periods of the most intense wind (Figure 12).

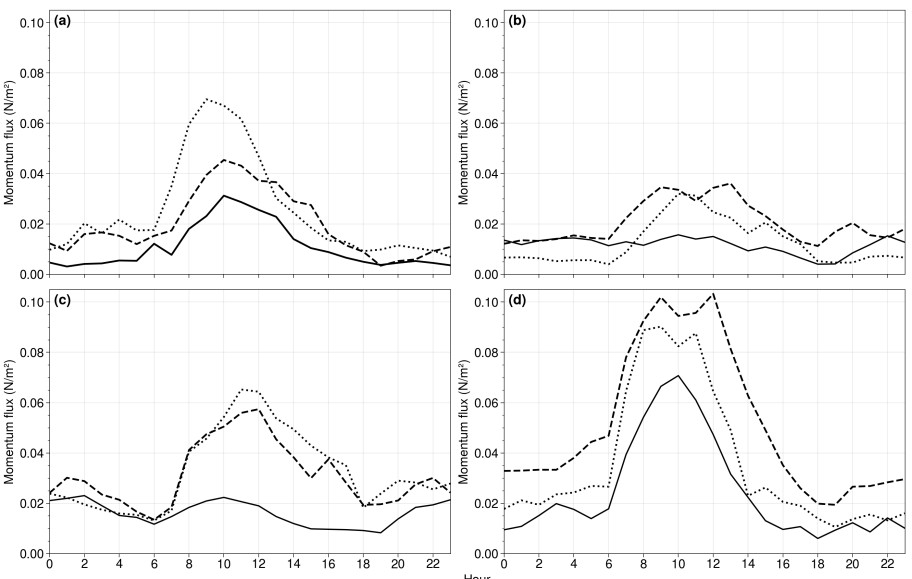

**Figure 12.** Diurnal cycle of momentum flux (N/m²) for (**a**) January, (**b**) April, (**c**) July, and (**d**) October. The normal year (dashed lines) is 2013, the dry year (dotted lines) is 2015, and the rainy year (solid lines) is 2020. Positive value is downward flux.

The maximum momentum flux reached 0.07 N/m² in the dry year, 0.045 N/m² in the normal year, and 0.025 N/m² in the rainy year (Figure 12a). In April, the rainy year presented the lowest momentum flux and the lowest variation during the studied period, with the minimum reaching 0.005 N/m² and the maximum reaching 0.015 N/m². These values indicate that, in April, during the rainy year, there was little mixing and few processes that generated turbulence at the water–atmosphere interface. In July, the dry and normal years showed slight similarities, although the maximum moment of flux in the dry year was higher than during the normal year (Figure 12c).

In October, as already shown in Figure 12d, the wind was more intense during the normal year. During this year, the wind presented higher momentum flux, reaching values greater than 0.1 N/m². This behavior indicates that, during October, in the rainy year, the surface layer of the lake experienced more turbulence caused by mechanical factors, and the surface and lake bottoms mixed well (Figure 12d).

### 3.5. Mixing Layer Depth

FLake was able to reproduce variation in the daily cycle for the depth of the mixing layer. During the day, the mixing layer was shallower than during the night. This difference in depth occurs because, during the day, there is increased heat flux upward out of the lake surface, creating a stratification where the surface is warmer and the thermocline is deeper. At night, on the other hand, heat fluxes decrease and the mixing is driven mainly by the momentum flux that continues and prevails over thermal processes, stirring the water and causing an increase in the mixed layer and a reduction in the thermocline ([25,26]).

In January, for the dry and normal years, the model reproduced the greatest depth of the mixing layer, reaching around 7.00 and 6.00 m in depth, respectively.

In the rainy month, however, due to weaker winds, there was less shear process and little mixing; consequently, a shallower mixing layer developed at night (Figure 13a).

In April, the simulations for the three years of the study showed slight similarities in the variation in the daily cycle for the mixing layer, with a maximum of 5.50 m for the normal and rainy years and 5.00 m for the dry year. The same results were simulated in July, during which the maximum depths were similar, about 6.50 m for all three years (Figure 13b,c). The model showed the maximum depth of the mixing layer in October; the magnitude of the wind was most intense during October for all three years, influencing the shear process and mechanical transfer of momentum to the deeper layers at night. The

depth reached 16.50 m in the dry year, 18.00 m in the normal year, and 15.50 m in the rainy year.

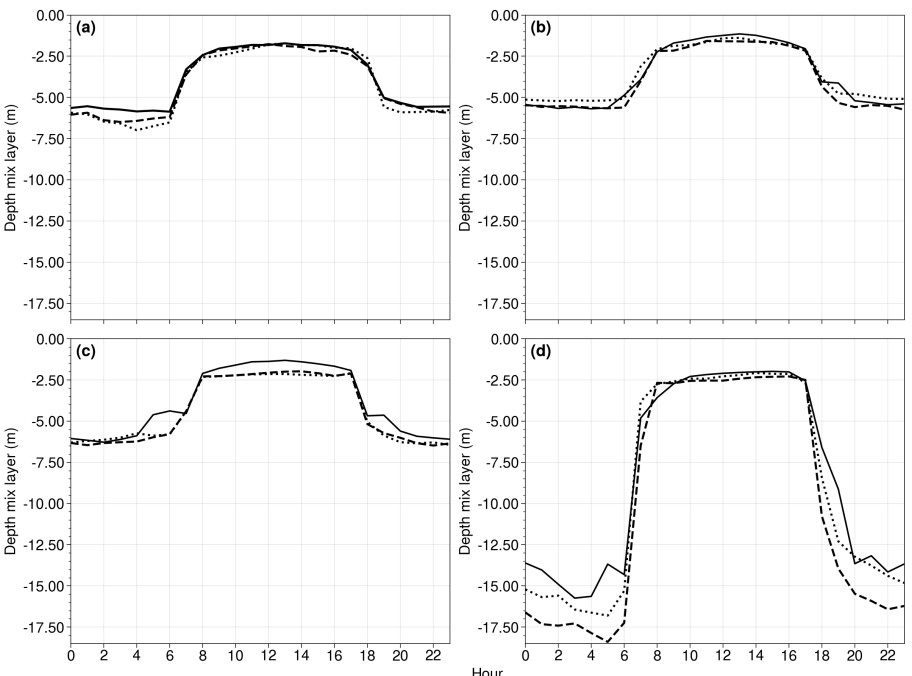

**Figure 13.** Diurnal cycle of the mixing layer depth (m) for (**a**) January, (**b**) April, (**c**) July, and (**d**) October. The normal year (dashed lines) is 2013, the dry year (dotted lines) is 2015, and the rainy year (solid lines) is 2020.

## 4. Discussion

Sobradinho Lake and its surroundings have a tropical climate characterized by distinct dry and rainy periods. The dry period extends from May to October, and the rainy period extends from November to April.

According to the climatology of 1991–2020, March had an average accumulated precipitation of about 364.90 mm, which differs from the previous climatology of 1981–2010, whose monthly average accumulated precipitation in March was 414.50 mm. The total annual precipitation shows a decreasing trend through the years. This variation in the region's precipitation regime directly impacts the energy and agricultural sectors, reducing the water volume in the reservoir and, consequently, reducing the amount of energy generated, as shown in Wijtkamp [27]. In agriculture, reduced precipitation can produce decreased crop productivity, leading to significantly lower yields and increased losses during harvest. Melo et al. [28] also observed a downward trend in precipitation totals, which has become more pronounced in the Sobradinho region since the 1990s.

These results do not indicate the causes for this decrease in precipitation volume over the last few decades. A more in-depth assessment of the dynamic patterns associated with the total precipitation reduction in this region is recommended.

The interannual variability from 2009 to 2020 highlighted 2015 as the dry year with the highest incidence of solar radiation at the surface and among the highest air temperatures. For the 10 m wind, the highest speeds were observed in 2013, a normal year for precipitation.

The mean 2 m air temperature peaked in October, and the minimum occurred in July. The monthly average maximum temperature was 28.60 °C, and the minimum was 25.40 °C. The maximum and minimum temperatures lagged by one month, corresponding to the maximum and minimum incoming solar radiation. These results were also observed in Santos [29].

The diurnal cycle of the lake surface temperature showed small amplitude variations in the four months studied. This small amplitude was due to the high heat retention capacity of water caused by its heat capacity, as described by Butcher [30].

In April, the model produced the largest thermal amplitude in the LST diurnal cycle during the rainy year, with 28.56 °C as the minimum and 29.12 °C as the maximum. This large amplitude occurred due to the weak wind intensity observed during the rainy year, which generated little turbulence near the surface, weakening the mixing process in the lake and, consequently, causing the surface layer to become warmer. In July and October, the lowest LST values were obtained in the months with the most intense winds, with maxima reaching 25.48 °C in July and 26.45 °C in October.

The 10 m wind greatly affected the fluxes. During the months with the strongest wind speeds, the model yielded greater variations in sensible heat fluxes, latent heat fluxes, and momentum fluxes. The wind, in addition to affecting the evaporation process of the lake, also helped in the mixing process, as the deeper layers became more strongly heated by the water coming from the surface layer. Thus, the mixing layer depth is strongly related to the wind speed. The mixing layer was deeper in July and October when 10 m winds were stronger.

## 5. Conclusions

Mean atmospheric conditions around the Sobradinho Lake were described in terms of interannual variability, mean annual cycle and mean diurnal cycle of meteorological variables, distinct rainy, dry and transition months, and typical rainy, dry, and normal years with respect to precipitation. These conditions were derived from the weather station located next to the lake. October, which is a transition month from dry to wet season, is the month of warmest temperatures, and strongest 10 m winds. The maximum incoming solar radiation at the surface and the strongest 10 m winds occur in September, thus preceding the temperature maximum by one month.

Generally, the FLake simulations satisfactorily reproduced Sobradinho Lake's thermodynamic variability. However, validating these simulations with observed data in the lake remains necessary. Therefore, future measurement campaigns in the lake are required to validate the corresponding models and improve our understanding of the physics and dynamics of the lake.

**Author Contributions:** E.O.A. carried out the numerical experiment with the Flake model, processing and manipulating data from automatic and conventional stations. S.C.C. contributed to writing the article and discussing the results. All authors have read and agreed to the published version of the manuscript.

**Funding:** This research was funded by Agência Nacional de Águas e Abastecimento and the Coordenação de Aperfeiçoamento de Pessoal de Nível Superior project ANA-CAPES No. 88881.144894/2017-01, and 88887.115869/2015-01, CAPES finance code 001, and Conselho Nacional de Desenvolvimento Científico e Tecnológico (CNPq) No. 312742/2021-5.

**Data Availability Statement:** Publicly available datasets were analyzed in this study. This data can be found here: https://portal.inmet.gov.br/ (accessed on 30 April 2023).

**Acknowledgments:** This work was funded jointly by the Agência Nacional de Águas e Abastecimento and the Coordenação de Aperfeiçoamento de Pessoal de Nível Superior project ANA-CAPES No. 88881.144894/2017-01, and 88887.115869/2015-01, and CAPES finance code 001. S.C. Chou thanks Conselho Nacional de Desenvolvimento Científico e Tecnológico (CNPq) for grant 312742/2021-5. The author thanks Dimitri Mironov for all contribution about FLake model information.

**Conflicts of Interest:** The authors declare no conflict of interest.

## Abbreviations

The following abbreviations are used in this manuscript:

| | |
|---|---|
| FLake | Freshwater Lake |
| CLM | Climate Limited-Area Modelling Community |
| DEPESCA/UFRPE | Fisheries Department of the Federal Rural University of Pernambuco |
| INMET | National Institute of Meteorology |
| ITCZ | Intertropical Convergence Zone |
| ASAS | South Atlantic Subtropical High |
| LST | Lake Surface Temperature |
| H | Sensible Heat Fluxes |
| LE | Latent Heat Fluxes |

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
