# Peer review of "Modeling the Effects of Local Atmospheric Conditions on the Thermodynamics of Sobradinho Lake, Northeast Brazil"

_climate, doi:10.3390/cli11100208_

Round 1
Reviewer 1 Report
The paper provides a detailed study of the climate and thermodynamic processes of the Sobradinho region, which is of significant importance for understanding the climate variability and ecological impact on the lake system in the area. Here are some suggestions to further enhance and improve the quality of the paper:
Introduction Section: In the introduction, it is recommended to provide a more detailed introduction to the significance of the Sobradinho region and the motivation behind the research.
Detailed Data Sources and Methods: When introducing the data sources and research methods, it would be beneficial to provide more detailed information, such as the locations of automatic weather stations, data collection frequencies, and operational parameters of the FLake model. This will aid other researchers in replicating your study, thereby increasing research reproducibility.
In-depth Result Interpretation: For instance, in the results section, delve into the analysis of trends in variables such as temperature and energy fluxes.
Enhance Discussion: Compare and discuss your research findings with existing literature. You can discuss the importance of these results for climate variability, ecosystem dynamics, and potential environmental impacts in the Sobradinho region.
Additionally, there are two minor issues that need to be addressed:
Maintain Consistent Precision: Throughout the entire paper, ensure that all values are presented with two significant digits.
Uniform Formatting of Units: Maintain a consistent format for units throughout the paper, specifically using "W/m2" instead of "Wm −2". This formatting should also be applied to figures and tables in the paper.
Author Response
Introduction Section: In the introduction, it is recommended to provide a more detailed introduction to the significance of the Sobradinho region and the motivation behind the research.
A: Recommendations were followed and the introduction was improved
Detailed Data Sources and Methods: When introducing the data sources and research methods, it would be beneficial to provide more detailed information, such as the locations of automatic weather stations, data collection frequencies, and operational parameters of the FLake model. This will aid other researchers in replicating your study, thereby increasing research reproducibility.
A: Thanks for the sugestion. The description of the methodology was improved, I tried to be more detailed in the way it was suggested.
In-depth Result Interpretation: For instance, in the results section, delve into the analysis of trends in variables such as temperature and energy fluxes.
A: Thanks for the sugestion. The analyzes were in-depth, including more details throughout the discussion of the results. However, no trend graphs were added.
Enhance Discussion: Compare and discuss your research findings with existing literature. You can discuss the importance of these results for climate variability, ecosystem dynamics, and potential environmental impacts in the Sobradinho region.
A: The recommendations were followed and the discussion of results was improved.
Additionally, there are two minor issues that need to be addressed:
Maintain Consistent Precision: Throughout the entire paper, ensure that all values are presented with two significant digits.
A: The suggested modifications were made.
Uniform Formatting of Units: Maintain a consistent format for units throughout the paper, specifically using "W/m2" instead of "Wm −2". This formatting should also be applied to figures and tables in the paper.
A: The suggested modifications were made.
Reviewer 2 Report
Dear Authors,
I think you should revise "references" before of presenting another time your paper,
Author Response
I think you should revise "references" before of presenting another time your paper,
A: References have been reviewed.
Reviewer 3 Report
This manuscript aims to study the climate variability and its impacts on the temperature of the Sobradinho Lake in Northeast Brazil. The results are helpful to understand climate variability around the Sobradinho Lake. However, the main conclusions of the current research need to be further organized and summarized, and the logical relationship between the various parts of the results needs to be strengthened. Some comments on the present manuscript are as follows.
1. The language expression needs further improvement, and the overall structure and logic of the manuscript also need to be improved. The overall framework of this study is not clear enough.
2. For the Abstract, the content of the abstract needs further refinement and improvement, and the main research results are not well reflected in the current abstract.
3. Introduction section: At present, there is only one paragraph. It is recommended to divide the content into different paragraphs according to logical relationships.
4. Line 80: “The years 2013, 2015, and 2020 were chosen as they correspond to normal, dry, and wet years, respectively, in Sobradinho Lake area.” The detailed method to choose the years should be introduced. Why only select one year for each type? It is better to select several typical years for each type to increase the statistical significance of the results. If it is only one year, it is too individual and the statistical significance is limited.
5. For the result section, what is the internal logical connection between the various research results in each sub-section (3.1 Interannual variability; 3.2. Atmospheric forcing; 3.3. Lake conditions; 3.4. Depth of the mixing layer)? I can not follow the logical relationship between different sub-sections. The overall approach and framework of this study are not clear enough.
6. Conclusion section: At present, there is only one paragraph. It is recommended to divide the content into different paragraphs according to logical relationships.
7. There are no discussions in the present manuscript. More discussions are needed to be added in the manuscript.
8. At present, there is a lack of sufficient mechanism analysis in research. For example, for the interannual variation, the analysis is shallow and there is no more in-depth specific analysis.
9. In this study, the climate variability involves different time-scales (interannual, seasonal, diurnal). It is necessary to arrange the content of different time-scales more logically in the manuscript.
10. There are multiple missing punctuation points in the main text. For example, Line 5: “revisited Observations”, Here, punctuation is required in the middle between “revisited” and “Observations”.
11. Line 20:” it was possible to observe negative values of H and in the nighttime”. Here, what does “H” represent?
The language expression needs further improvement, and the overall structure and logic of the manuscript also need to be improved.
Author Response
1. The language expression needs further improvement, and the overall structure and logic of the manuscript also need to be improved. The overall framework of this study is not clear enough.
A: Thanks for the recommendation. The general structure of the work was adjusted.
2. For the Abstract, the content of the abstract needs further refinement and improvement, and the main research results are not well reflected in the current abstract.
A: Thank you very much for the suggestion. The work summary has been improved.
3. Introduction section: At present, there is only one paragraph. It is recommended to divide the content into different paragraphs according to logical relationships.
A: Thanks for the recommendation. The introduction paragraphs have been divided.
4. Line 80: “The years 2013, 2015, and 2020 were chosen as they correspond to normal, dry, and wet years, respectively, in Sobradinho Lake area.” The detailed method to choose the years should be introduced. Why only select one year for each type? It is better to select several typical years for each type to increase the statistical significance of the results. If it is only one year, it is too individual and the statistical significance is limited.
A: Thanks for the recommendation. The methodology was detailed and the reason for choosing normal, dry and rainy years was specified. However, several years were not selected because the automatic station does not have records of several periods with data on air temperature, solar radiation and wind magnitude.
5. For the result section, what is the internal logical connection between the various research results in each sub-section (3.1 Interannual variability; 3.2. Atmospheric forcing; 3.3. Lake conditions; 3.4. Depth of the mixing layer)? I can not follow the logical relationship between different sub-sections. The overall approach and framework of this study are not clear enough.
A: Thanks for the notes. The sequence and connection between the results have already been justified.
6. Conclusion section: At present, there is only one paragraph. It is recommended to divide the content into different paragraphs according to logical relationships.
A: Thank you very much for the recommendation. Paragraphs have been added in the conclusion.
7. There are no discussions in the present manuscript. More discussions are needed to be added in the manuscript.
A: Thanks for the recommendation. A discussion session has been added.
8. At present, there is a lack of sufficient mechanism analysis in research. For example, for the interannual variation, the analysis is shallow and there is no more in-depth specific analysis.
A: Thanks for the recommendation. The analyzes were in-depth as recommended.
9. In this study, the climate variability involves different time-scales (interannual, seasonal, diurnal). It is necessary to arrange the content of different time-scales more logically in the manuscript.
A: Thanks for the recommendation. They were rewritten according to the recommendations.
10. There are multiple missing punctuation points in the main text. For example, Line 5: “revisited Observations”, Here, punctuation is required in the middle between “revisited” and “Observations”.
A: Thanks for the comments. The recommendations were followed.
11. Line 20:” it was possible to observe negative values of H and in the nighttime”. Here, what does “H” represent?
A: H represents sensible heat flux
Round 2
Reviewer 2 Report
Manuscript has been improved. But, I think, a final path will be needed: order the references in the text
Author Response
Manuscript has been improved. But, I think, a final path will be needed: order the references in the text.
A: Thank you for your recommendation. References were ordered
Reviewer 3 Report
1. Line 19:” Flake simulated negative values of H”. The reader does not know what H refers to, if there is no relevant explanation in the previous text.
2. “Observations showed a reduction in rainfall during 1991-2020 compared to 1981-2010.” Why do the two selected periods (1991-2020; 1981-2010) overlap, rather than two periods that do not overlap before and after?
3. Line 121: “Figure 3 shows the average taken from two periods, one from 1981 to 2010 and the other from 1991 to 2020.” Actually, the differences between these two periods are not interannual variability (It is decadal variability). 3.1. Interannual variability, section 3.1 should more focus on interannual variability.
4. How to make modifications should be explained in detail and marked in the text.
Minor editing of English language required.
Author Response
- Line 19:” Flake simulated negative values of H”. The reader does not know what H refers to, if there is no relevant explanation in the previous text.
A: The variable H is the sensible heat flux; the symbol is now defined in line 17.
- “Observations showed a reduction in rainfall during 1991-2020 compared to 1981-2010.” Why do the two selected periods (1991-2020; 1981-2010) overlap, rather than two periods that do not overlap before and after?
A: The periods overlap, but we intend to show that despite the overlap of some years, the climatology shows a change in statistics (Maximum and Minimum), signaling local climate changes. It was clarified on line 148.
- Line 121: “Figure 3 shows the average taken from two periods, one from 1981 to 2010 and the other from 1991 to 2020.” Actually, the differences between these two periods are not interannual variability (It is decadal variability). 3.1. Interannual variability, section 3.1 should more focus on interannual variability.
A: In this section, the intention is to highlight the rainy months, transition months, and dry months. A new section 3.2 annual cycle has been added.
- How to make modifications should be explained in detail and marked in the text.
A: Modified texts are highlighted in blue. Thank you for your recommendation.